# Opioid-Free Anesthesia in Bariatric Surgery: Is It the One and Only? A Comprehensive Review of the Current Literature

**DOI:** 10.3390/healthcare12111094

**Published:** 2024-05-27

**Authors:** Piotr Mieszczański, Marcin Kołacz, Janusz Trzebicki

**Affiliations:** 1st Department of Anaesthesiology and Intensive Care, Medical University of Warsaw, Lindleya 4 Str., 02-005 Warsaw, Poland; marcin.kolacz@wum.edu.pl (M.K.); janusz.trzebicki@wum.edu.pl (J.T.)

**Keywords:** opioid-free anesthesia, bariatric surgery, multimodal analgesia, pain treatment, dexmedetomidine, lidocaine, ketamine, nociception, anesthesiology

## Abstract

Opioid-free anesthesia (OFA) is a heterogeneous group of general anesthesia techniques in which the intraoperative use of opioids is eliminated. This strategy aims to decrease the risk of complications and improve the patient’s safety and comfort. Such potential advantages are particularly beneficial for selected groups of patients, among them obese patients undergoing laparoscopic bariatric surgery. Opioids have been traditionally used as an element of balanced anesthesia, and replacing them requires using a combination of coanalgesics and various types of local and regional anesthesia, which also have their side effects, limitations, and potential disadvantages. Moreover, despite the growing amount of evidence, the empirical data on the superiority of OFA compared to standard anesthesia with multimodal analgesia are contradictory, and potential benefits in many studies are being questioned. Additionally, little is known about the long-term sequelae of such a strategy. Considering the above-mentioned issues, this study aims to present the potential benefits, risks, and difficulties of implementing OFA in bariatric surgery, considering the current state of knowledge and literature.

## 1. Introduction

Currently, approximately 2.5 billion adults in the world are overweight, of which 890 million are obese, and this number is constantly increasing [1]. Approximately 600,000 patients undergo bariatric surgery annually; a much larger group has indications for it [2]. Markedly, it has been reported that almost 75% of patients undergoing laparoscopic bariatric surgery may experience moderate and severe pain [3]. The most potent group of pharmacological substances used to alleviate it and to surpass the nociception intraoperatively during general anesthesia are opioids, and it is estimated they are administered to approximately 99% of patients in the USA perioperatively [4,5]. Their mechanism of action affects the modulation of pain impulses, mainly in the central nervous system, making them highly effective. Unfortunately, despite their high analgesic potency, opioid use is connected with the risk of adverse reactions in the postoperative period, such as oversedation, respiratory depression, postoperative nausea and vomiting (PONV), as well as opioid hyperalgesia. These side effects attributed to opioids complicate the postoperative period, increase costs, and prolong the length of hospital stays [6].

Considering the specificity of patients with obesity undergoing bariatric surgery and their increased risk of complications in the intra- and postoperative period, new methods of anesthesia and analgesia are constantly being searched for, which would allow for greater safety and comfort. One of the techniques used for this purpose is opioid-free anesthesia (OFA), but despite growing evidence, its use remains controversial, and its benefits are questioned.

## 2. Purpose

This article aims to critically evaluate published material on opioid-free anesthesia in bariatric surgery, providing an integrated and synthesized overview of the current state of knowledge. In our review, we plan to assess whether the opioid-free anesthesia technique has an advantage over anesthesia with multimodal analgesia, including opioid use, which is currently standard in bariatric surgery.

## 3. Material and Methods

The literature search was conducted using the PubMed, Web of Science, and SCOPUS databases using the keywords “opioid free anesthesia”, “opioid free analgesia” and “bariatric surgery anesthesia” in the time period from inception to April 2024. A manual search of the reference lists of the selected publications was also performed to identify additional studies for potential inclusion. The initial data search yielded 1318 papers from PubMed, 2847 from Web of Science, and 2457 from SCOPUS. A total of 6622 articles were identified, and after removing duplicates using Rayyan (Johnson & Phillips, 2018, Newport, UK), the remaining 2040 were screened for eligibility. We have included randomized controlled trials referring to bariatric surgery published in English. Through a title and abstract review, 54 studies were further examined, of which 13 studies were retrieved. We excluded 2 studies as they were referring to open bariatric surgery, i.e., laparotomy, which is currently obsolete. The selection process is depicted in Figure 1. After a full text article assessment, 11 relevant randomized controlled trials were included in this review. The relevant studies are depicted in Table 1.

## 4. Results

### 4.1. Opioid-Free Anesthesia—Definition and Assumptions

According to most definitions, OFA is a heterogeneous group of techniques that include general anesthesia without systemic or regional opioid administration [18]. By an alternative definition, OFA involves the use of various methods to eliminate opioids and avoid their side effects without negatively affecting the patient’s comfort [19].

Such methods include regional anesthesia (RA) techniques. In the setting of modern laparoscopic bariatric surgery, RA can be used as an element of anesthesia with multimodal analgesia or OFA. The simplest use of RA is the infiltration of the surgical site with local anesthetics [20]. Moreover, interfascial plane blocks, such as Transversus Abdominis Plane (TAP) Block [8,21], Erector Spinae Plane (ESP) Block [22], or the intraperitoneal administration of local anesthetics [23], including a promising blockade of the autonomic innervation of the stomach [24] when used as a part of multimodal analgesia, are also effective in perioperative opioid-sparing. Referring to neuraxial blockades, although epidural and even combined thoracic spinal-epidural anesthesia have been successfully implemented for laparoscopic bariatric procedures [25,26,27], they are deemed to be too invasive and prevent early mobilization, which has a priority in ERAS strategy [28]. On that basis, there is a consensus that this form of RA can be considered nowadays only in rare cases of open bariatric surgery, but not laparoscopic [28]. Additionally, non-opioid analgesics and coanalgesics (Table 2), including drugs to prevent hyperalgesia and non-pharmacological agents, are also used in OFA.

For practical reasons, a clear distinction should be made between anesthesia without intraoperative opioids and opioid-free postoperative analgesia. Articles reporting the complete elimination of opioids both during and after bariatric surgery [29,30] are based on single case reports and not on routine, reliable practice. In the OFA concept, it is crucial to distinguish between pain and nociception. Pain, according to the definition of IASP (International Association for the Study of Pain), is an unpleasant sensory and emotional experience that assumes a state of consciousness [31]. Nociception, conversely, refers to a stimulus’s reception and excitation transmission in the nervous system [32]. During general anesthesia, the patient’s pain perception is disabled, and nociception is based on the response of the sympathetic nervous system, which is mainly the easiest to assess the parameters of the circulatory system, such as heart rate (HR) or blood pressure (BP) [33]. OFA assumes the suppression of the sympathetic nervous system in response to a pain stimulus and the modulation of nociception through the use of methods other than the administration of opioids [34]. Such methods include the use of drugs from the alpha 2 agonists group, lidocaine, ketamine, magnesium sulfate, beta-blockers, or gabapentinoids (Table 2) [35]. Despite its potential benefits, especially for obese patients undergoing bariatric surgery, OFA is controversial and is not currently recommended as a standard treatment. This is due to the limited scientific evidence and frequent reliance on expert opinion [36,37]. This article aims to present the potential benefits, risks, and difficulties associated with this technique.

### 4.2. Potential Benefits of OFA in Bariatric Surgery

Opioid side effects make it difficult to mobilize the patient early, which is a priority in modern bariatric surgery, in which laparoscopic sleeve gastrectomy (LSG) and Roux-en-Y gastric bypass (LGB) are the most frequently performed operations [38]. The use of a comprehensive perioperative care protocol to improve the outcomes of surgical treatment (ERAS, Enhanced Recovery After Surgery), including the reduction or elimination of opioid therapy, is of fundamental importance for patient safety and comfort [28]. Based on its assumptions, the OFA technique should reduce the incidence of respiratory complications, excessive sedation, and postoperative nausea and vomiting (PONV). It should also ensure comparable or better pain control and avoid the risk of opioid-induced hyperalgesia (OIH) [19]. On the other hand, there are concerns about the stability of the circulatory system in patients anesthetized with this technique, as well as whether OFA actually blocks pain conduction or only the stimulation of the sympathetic nervous system in response to a stimulus, and what the long-term consequences of such technique may be [36].

### 4.3. Respiratory Complications and Oversedation

The avoidance of opioid-induced respiratory depression (OIRD) and oversedation in the postoperative period, and thus an improvement in patient safety, are the main reasons for the interest in OFA in bariatric anesthesia [39,40,41]. The increased susceptibility to respiratory complications in obese patients involves numerous factors such as reduced functional residual capacity (FRC), atelectasis, air leak, co-occurrence of obstructive sleep apnea (OSA), obese hypoventilation syndrome (OHS) and pulmonary hypertension [42,43,44]. An increased risk of respiratory complications is common after the administration of opioids and may be clinically significant even without clear signs of overdose [45,46]. In a meta-analysis devoted to the factors of OIRD, one of the conclusions is the possibility of reducing the risk by using opioid-sparing techniques [45]. In line with this recommendation, the Enhanced Recovery After Bariatric Surgery (ERABS) guidelines suggest the standard use of multimodal analgesia involving coanalgesics, regional anesthesia techniques, and non-opioid analgesics [28]. Despite the potentially improved safety due to the complete elimination of opioids, the amount of evidence for the greater safety of OFA in bariatric surgery compared to general anesthesia with multimodal analgesia is minimal. Such evidence is provided by the study of Mulier et al., in which the rate of desaturation < 94% in the postoperative period in patients in the opioid group was 50%, while in only 2 of 23 in the OFA group [9]. However, the issue remains controversial as other trials do not confirm such an effect [13,14,17]. These discrepancies may be explained by the sedative effect of some drugs used in OFA, especially dexmedetomidine, which may have an ambiguous impact on convalescence and the possibility of early mobilization after surgery, presumably dose-dependent. This effect is caused by a presynaptic effect on alpha 2 receptors in the locus coeruleus [47]. As far as the impact of OFA on recovery is concerned, the trial’s results are inconsistent. In a study dedicated to bariatric surgery, dexmedetomidine was associated with meeting the discharge criteria faster in the recovery room; however, it was not associated with a reduction in the length of the hospital stay [48]. Similar results were obtained by Ulbing et al., in whose study patients in the OFA group achieved a statistically higher result in the subjective assessment of recovery using the QoR-40 form 24 and 48 h after surgery, but this was not associated with a shortened hospital stay [15]. On the contrary, in the trial by Clanet et al., no statistically significant difference between the QoR-40 questionnaire results was obtained 24 and 30 days after the surgery [13]. Furthermore, recovery after surgery can also be affected by hallucinations, especially as ketamine is frequently used in OFA protocols. In one study, they were reported in up to 7% of patients; despite this side effect, the satisfaction level of patients during the perioperative period was not affected [12].

### 4.4. Postoperative Nausea and Vomiting

PONV has a multifactorial etiology and significantly contributes to the diminished comfort of patients undergoing laparoscopic bariatric surgery [9] as well as being the most frequent cause of the readmission of patients after bariatric surgery [49]. PONV may also hinder the patient’s rapid mobilization and pose a risk of increased blood pressure, wound dehiscence, or bleeding in the perioperative period [50]. Opioid use is one of the few modifiable factors of PONV, especially as its incidence is dose-dependent [51]. There is clear evidence of a reduction in the prevalence of this complication in patients undergoing laparoscopic bariatric surgery, demonstrated in prospective trials [7,9,11,13,14,15] and in a meta-analysis dedicated to this group of patients in comparison to the anesthesia with multimodal analgesia group [52]. Still, there are discrepancies regarding the duration of the beneficial effect. In a prospective, randomized study by Ziemann-Gimmel et al. [11], researchers demonstrated that Total Intravenous Anesthesia (TIVA) OFA allows for a more significant reduction of the risk of PONV than triple antiemetic prophylaxis and the reference point was the frequency and severity of PONV 24 h after surgery [11]. A similar beneficial effect was maintained in the study by Mulier et al. [9], but in other trials, this effect was shown only in the immediate hours after surgery [13,14]. In the latter trial, in the OFA group, significantly fewer patients required antiemetics, but lower PONV incidence persisted until the 4th postoperative hour. In other studies, however, the observed differences did not reach statistical significance [10,17].

### 4.5. Pain Control and Reduction of Postoperative Opioid Consumption

Publications supporting multimodal analgesia in patients undergoing bariatric surgery, as included in the ERAS Society guidelines [28], demonstrate a reduction in the doses of opioids required for pain treatment owing to the use of coanalgesics from the alpha 2 agonists group [53,54,55], lidocaine [56], magnesium sulfate [57] or ketamine [58]. However, there is limited research comparing OFA and anesthesia with multimodal analgesia, including opioid use, for postoperative pain management and opioid dosage, and existing results are inconclusive. In a meta-analysis dedicated to bariatric surgery by Hung et al., a statistically significant reduction in the NRS score was demonstrated in OFA group patients 24 h after surgery; however, considering that this difference did not exceed 1 point on the NRS scale, its clinical significance is questionable [59]. The same study did not reveal a reduction in the total dose of opioids administered postoperatively, only in the initial period in the recovery room [59]. A study by Menck et al. demonstrated no statistically significant differences in both pain scores and opioid requirements at any given time point [17]. Mulier’s 2018 prospective, randomized trial stands out as one of the most notable trials highlighting OFA’s benefits. The study involved 50 patients who were assigned to either the OFA group or the opioid administration group. The OFA group required significantly less opioids in the recovery room, 4.9 to 15.3 mg of morphine (*p* = 0.04), and had a lower VAS score of 1.7 to 4.9 (*p* = 0.01) [9]. The study presented some limitations in terms of incomplete information regarding the duration of patients’ stay in the PACU before their transfer to the ward, where there was no significant difference in opioid consumption, and the OFA group presented with only a slightly better VAS score of 2.0 compared to 3.3 in the opioid group (*p* = 0.016) [9]. Similar results were obtained by Ahmed et al. in their randomized controlled trial, in which lower pain scores were noted 4 and 6 h after the surgery, whereas the total morphine consumption was statistically but not clinically lower: 5.8 in the OFA group vs. 7.2 mg in the opioid group (*p* = 0.003). [7]. In line with Mulier’s trial, a reduction in the dose of opioids only in the initial postoperative period was also demonstrated by Mieszczański et al. [14]. A possible explanation for this difference may be the fact that drugs used in OFA have a half-life of several hours, and their effects subside shortly after the cessation of their infusion [56,60]. One of the few studies with the continuation of the coanalgesics in the postoperative period is the work of Ulbing et al., which demonstrated a lower opioid consumption and a lower VAS pain score in the OFA group [15]. Maintaining the infusion of these drugs may be critical to achieving the clinical significance of the benefits of OFA in bariatric procedures. This has also been demonstrated in a case report of a patient undergoing LSG as a bridge to eligibility for lung transplantation for interstitial disease [61]. In that case, the infusion of coanalgesics was maintained for 24 h after surgery, which resulted in low total opioid consumption and acceptable NRS scores. In conclusion, there is still a lack of unequivocal evidence that OFA is associated with comparable or better-quality pain management. While the beneficial effects of OFA can be prolonged with the continued administration of coanalgesics in the postoperative period, the assessment of the significance of the clinical benefit of this approach requires further research.

### 4.6. Tolerance, Opioid-Induced Hyperalgesia (OIH), and Long-Term Sequelae

Intraoperative administration of opioids may spark the development of their acute tolerance, which is associated with an increase in total opioid requirements and increases the frequency of side effects [62,63]. Another coexisting problem is the occurrence of OIH, which entails a lower pain threshold and allodynia [64]. This phenomenon has a separate pathophysiology, but the clinical manifestations are similar and, in practice, difficult to differentiate [64]. OIH and the acute development of tolerance are crucial in bariatric surgery, as frequently used remifentanil has the greatest potential in this respect [65], especially at a high dose [66]. This may have an impact on postoperative pain management [55], and the relationship appears to be dose-dependent [65]. When it comes to proven effects in the prevention and treatment of OIH and the development of tolerance to opioids, ketamine or magnesium sulfate (NMDA receptor antagonists), as well as alpha 2 agonists, often included in OFA protocols, are used [67,68,69,70]. OFA has a beneficial effect on reducing the doses of opioids used and, therefore, on the prevention of OIH and the development of tolerance to opioids. This is of significant importance, as there exist publications, including a meta-analysis, that indicate the lack of any discernible benefits associated with the administration of opioids prior to the pain stimulus [71]. Another question is the impact of OFA on the incidence of Persistent Postoperative Pain, which is estimated to occur in up to 30% of patients [72], with the incidence ranging in the literature from 5 to 54.4% [73,74]. Considering the potential detrimental effects associated with the use of opioids, the implementation of OFA may serve as a viable solution for mitigating such risks. Additional research is necessary considering the limited availability of only one small study that does not indicate significant differences in chronic pain incidence or intensity among patients who underwent hysterectomy over a six-month duration [75].

### 4.7. Hemodynamic Stability

Laparoscopic bariatric surgery is associated with the risk of hemodynamic instability, which is caused by the insufflation of the peritoneal cavity with carbon dioxide and an increase in intra-abdominal pressure, positioning a patient in a steep anti-Trendelenburg position, or from co-occurring cardiovascular diseases in obese people [76]. In this respect, there are concerns about the use of OFA in this group of patients, which is due to the depressive effect of the drugs used on the circulatory system. Dexmedetomidine, the most commonly used alpha 2 agonist, has a parasympathomimetic effect on the cardiac conduction system and inhibits the sympathetic component of the cardiac plexus, resulting in bradycardia and even sinoatrial block. Moreover, this drug causes the relaxation of the vascular tonus and a decrease in systemic vascular resistance (SVR), leading to a decrease in BP (Blood Pressure). On the other hand, with rapid administration, paradoxical vasoconstriction with an increase in BP may occur due to the non-specific stimulation of alpha 1 receptors [60]. With regard to these dose-related adverse effects of dexmedetomidine, the large multicenter POFA study, which compared OFA anesthesia with dexmedetomidine to anesthesia with remifentanil, demonstrated adverse effects of the first technique, such as hemodynamic instability in the form of bradycardia and in one case, asystole, as well as a more frequent occurrence of hypoxia and greater sedation of patients in the postoperative period [77]. The study was interrupted for safety reasons. The article was criticized because the use of high doses of dexmedetomidine (an average of 1.2 mcg/kg/h) with a long anesthesia time (an average of 268 min) raises doubts, and what is more, the limitation is the high heterogeneity of the anesthetic procedures, of which bariatric surgeries consist only a small portion. Nevertheless, POFA is a significant prospective study that questions the safety and soundness of the OFA technique [61]. Lidocaine also has a hypotensive effect through its negative inotropic effect on cardiomyocytes, similar to magnesium sulfate, an NMDA receptor antagonist, or an antagonist of beta 1 receptors, such as esmolol [78,79,80]. Lidocaine at a dose of 1.5 mg/kg can even be used for controlled hypotension during general anesthesia [78]. The effect of ketamine is ambiguous, although increasing the tension of the sympathetic nervous system causes an increase in HR, BP, and SV (Stroke Volume) while maintaining SVR. Still, in some patients, it may have a negative inotropic effect, causing hypotension and bradycardia [81]. Taking into account the effects mentioned above, hypotension when using OFA may be a problem and require the use of sympathomimetics such as ephedrine or catecholamines more frequently and in higher doses than in conventional anesthesia with multimodal analgesia, as well as more aggressive fluid therapy [14]. This may pose a risk, especially for patients with ischemic heart disease, hypovolemia, or orthostatic hypotension, and OFA is relatively contraindicated in this group [33]. Another problem is the intraoperative, low controllability of some drugs used in OFA, including alpha 2 agonists and lidocaine, the effects of which may be prolonged due to their half-life [60,82]. In the case of hypotension and bradycardia, even after the discontinuation of the administration of these drugs, the disappearance of their effects will be delayed. On the other hand, the data from the clinical trials are inconclusive and contradictory, as in another recent trial, the differences between OFA and remifentanil groups in terms of hemodynamic stability have not reached statistical significance [13], and patients in the opioid group received more fluids than anesthetized without opioids. Similar results, with no significant differences in hemodynamic parameters between OFA and opioid groups, were obtained in a study by Mansour et al. In this particular study, however, ketamine was the only coanalgesic utilized, avoiding the potential hypotension associated with lidocaine, alpha 2 agonists, or magnesium sulfate use [12]. Therefore, further studies are required on the impact of OFA on hemodynamic stability, and the observed differences may result from the heterogeneity of utilized OFA protocols, especially in proportions of particular coanalgesics used.

### 4.8. Intraoperative Nociception and Monitoring

Repeated nociceptive stimulation reaching higher levels of the nervous system causes central sensitization, defined by IASP as increased responsiveness of nociceptive neurons in the central nervous system to their normal or subthreshold input [31]. This process contributes to the development of acute and persistent postoperative pain [34,83], which, inadequately treated, apart from numerous other unfavorable effects, is one of the main factors in its chronification. The adopted intraoperative opioid dosage is, in most cases, based on the features of sympathetic nervous system stimulation and is considered an indicator of nociception, i.e., based on the assessment of hemodynamic parameters [35]. Drugs used in OFA are weak analgesics (alpha 2 agonists, ketamine, lidocaine) or have no direct analgesic effect. Therefore, it is unclear whether these drugs provide hemodynamic stability by effectively attenuating nociception or simply blocking the effector, namely the sympathetic nervous system, and what the consequences of this might be. Monitoring intraoperative nociception is challenging and includes the assessment of the vegetative system based on heart rate variation (HRV), the Analgesia Nociception Index (ANI), the NoL Index [84] or the High-Frequency Variability Index (HFVI) [85] as well as the measurement of pupil width [86], and indirectly by determining stress hormones before and after surgery (e.g., cortisol) [9]. Parameters derived from the EEG recording (e.g., bispectral index BIS, entropy) are not suitable for strictly assessing nociception, and their assessment after ketamine administration is unreliable [87]. The amount of research on obese patients anesthetized using the OFA technique is scarce. The literature describes two case reports of such patients with class III obesity and nociception monitoring by ANI assessment [29,88]. In both articles, this technique allowed for maintaining the ANI in the desired range of 50–70, which allows for reasonable control of nociception and minimizing the risk of the patient feeling pain after waking up [29,88]. However, this method, like other techniques based on HRV assessment, is subject to significant limitations, including the use of atropine, sympathomimetics, or other drugs affecting HR used in OFA [89,90]. In the previously cited work, Mulier demonstrated lower cortisol concentration in patients anesthetized for laparoscopic bariatric surgery in the OFA group compared to anesthesia with sufentanil administration. The cortisol concentration was measured before the induction of anesthesia and then in the postoperative department; the increase was statistically significantly lower in the OFA group, which may indirectly indicate lower perioperative stress in this group of subjects [9]. Moreover, a beneficial effect of OFA on reducing immunologically mediated stress response was demonstrated in a study by Campos-Perez et al. [16], in which patients in the OFA group undergoing LGB had lower interleukin 6 (IL6) serum concentrations postoperatively 13 pg/mL (5.43–22) vs. 49.58 pg/mL (18.50–112.20), respectively, *p* = 0.019. IL-6 is considered to be one of the most important pro-inflammatory interleukins and biomarkers of inflammation and immune activation. On the other hand, no differences in other primary outcomes, TNF-α and IL-1β serum concentration, were detected. In this study, no clinical or statistical differences in parameters such as PONV incidence or NRS scale result were observed. Due to observation time being limited to 24 h after the surgery, no conclusions on the clinical significance of IL-6 reduction sequelae can be made [16].

Considering the limited data on blocking nociception and stress response using the OFA technique and the long-term consequences of this method of anesthesia, more research is warranted in this area.

## 5. Discussion

The literature proves that OFA is feasible and can be successfully implemented as a strategy for bariatric surgery [19]. As far as certainties are concerned, there is unequivocal evidence that OFA decreases the incidence and severity of PONV compared not only to opioid-liberal anesthesia but also to opioid-sparing [11,52]. In other fields, in which OFA is expected to be superior, there is conflicting or scarce evidence, or the evidence is not allowed to be adapted as a standard, and this evidence refers to improving postoperative pain management and decreasing the postoperative opioid requirements [7,9,10,12,13,14,15,17], the incidence of opioid-induced respiratory depression or oversedation [9,13,14,17], maintaining hemodynamical stability [12,13,14] or finally, improving the recovery [8,9,10,13,15] and long-term outcomes in terms of postoperative chronic pain incidence. One of the main factors contributing to this fact is the vast heterogeneity of OFA protocols and coanalgesics used, and even the adapted dosing regimens—ideal body weight, lean body weight, or adjusted body weight—as well as if and what type of RA techniques were used.

In our experience, OFA is associated with more interventions of the anesthetist intraoperatively, may pose a risk of hemodynamic instability, and does not shorten the length of hospital stays as compared with anesthesia with multimodal analgesia [14]. In our center, we use it in patients with the paramount risk of respiratory complications, for example, on chronic oxygen treatment [61,91] or with super-obesity (BMI > 50) in an individual risk assessment, and also in cases of severe PONV history. In such cases, to maximize the potential benefits, we maintain coanalgesic administration throughout up to 12–16 h after the surgery.

Future research concerning OFA should compare it with multimodal, low-opioid strategies (not just “opioid-based” anesthesia), assess the optimal dosing regimen and if prolonging coanalgesic administration postoperatively would improve the results, and finally, the long-term impact on the recovery and the postoperative pain chronification.

## 6. Conclusions

The elimination of opioids during anesthesia is possible, but it poses many difficulties and must be considered in the context of the entire perioperative period, not just the operating room. While, currently, adapting opioid-sparing strategies as an element of anesthesia with multimodal analgesia to minimize the use of opioids is considered a standard in bariatric perioperative care, there are indications that a more radical approach, such as OFA, may have advantages. Based on these assumptions, more and more centers are introducing their use. On the other hand, there is growing evidence that the benefits may be limited, and issues related to safety, long-term effects, and the place of OFA in the Fast Track Surgery doctrine are not resolved. Therefore, considering the meager amount of literature, OFA should not be used as a standard and only as an anesthetic technique in bariatric surgery. The question of the broader application of this method requires further research.

## Figures and Tables

**Figure 1 healthcare-12-01094-f001:**
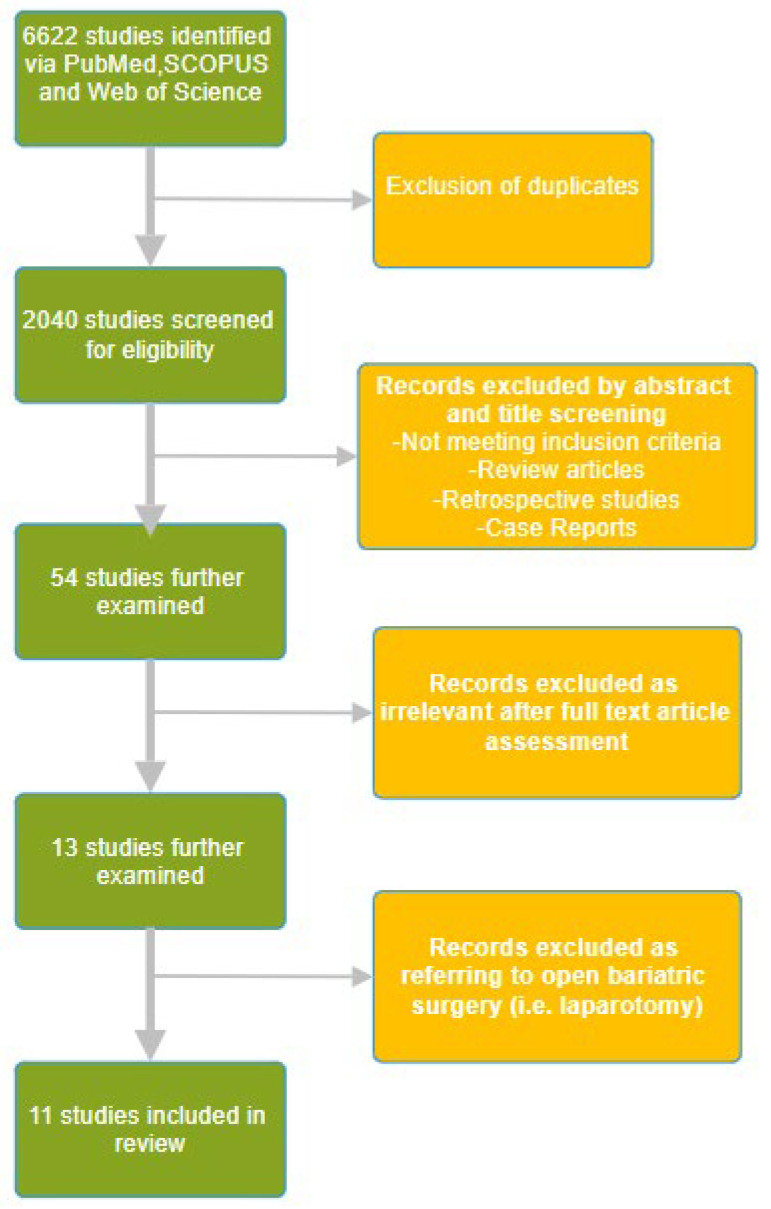
The randomized controlled trials selection flow diagram.

**Table 1 healthcare-12-01094-t001:** OFA randomized controlled trials included in the review. LSG—laparoscopic sleeve gastrectomy, LGB—laparoscopic gastric bypass, SADI-S—single anastomosis dueodeno-ileal bypass.

Author	Year	Number of Patients Enrolled	Type of Surgery	Coanalgesics Used	Primary Outcome Measure	Main Results
Ahmed SA et al. [7]	2022	80	LSG, LGB	Ketamine, Dexmedetomidine, Magnesium sulfate, Lidocaine	Morphine consumption in 24 h	OFA 5.8 vs. 7.2 mg, (*p* = 0.003)
Ibrahim M et al. [8]	2022	103	LSG	Ketamine, Dexmedetomidine, Lidocaine, Dexamethasone	Quality of recovery assessed by QoR-40, at the 6th and the 24th postoperative hour	At the 6th hour, the QoR-40 OFA median [IQR] was 180 [173–195] vs. 185 [173–191], (*p* < 0.0001), but no difference was found at the 24th hour (median values = 191 in both groups)
Mulier JP et al. [9]	2018	45	LSG, LGB	Ketamine, Dexmedetomidine, Lidocaine,	Non-specified	VAS score in the ward OFA group 2.0 vs. 3.3 *p* = 0.016, total morphine consumption 14.7 vs. 18.2 mg *p* = 0.33
Soudi AM et al. [10]	2022	60	Laparoscopic bariatric surgery	Ketamine, Dexmedetomidine, Lidocaine	Non-specified	Nalbuphine consumption in OFA 8.17 ± 4.8 vs. 23.67 ± 4.5 mg < 0.001
Ziemann-Gimmel P et al. [11]	2014	119	LSG, LGB, Laparoscopic gastric band	Ketamine, Dexmedetomidine	PONV incidence	OFA group 12 patients (20%) vs. 22 patients (37.3%) [*p* = 0.04; risk 1.27 (1.01–1.61)]
Mansour et al. [12]	2013	28	LSG	Dexamethasone, Ketamine	Heart rate, systolic, diastolic, and mean arterial blood pressure on induction and ½ hourly thereafter	No statistically significant differences between the groups
Clanet et al. [13]	2024	172	LGB	Dexamethasone, Ketamine, Magnesium sulfate, Dexmedetomidine, Lidocaine	Morphine consumption in 24 h	OFA 16 [13–26] vs. 15 [10–24] mg, (*p* = 0.183)
Mieszczański et al. [14]	2023	59	LSG	Dexamethasone, Ketamine, Magnesium Sulfate, Dexmedetomidine, Lidocaine	Oxycodone consumption at 1,6,12 and 24 h, pain scores at 1,6,12 and 24 h	OFA 1 h 4.6 mg vs. 7.72 mg (*p* = 0.008)
Ulbing et al. [15]	2023	99	LGB, LSG, Laparoscopic omega loop bypass, SADI-S	S-ketamine, Dexmedetomidine, Lidocaine, Magnesium Sulfate	Difference in the VAS within the first 24 h after surgery	OFA 2.2 [1–4.4] vs. 4.1 [2–6.5] *p* ≤ 0.001
Campos-Pérez et al. [16]	2022	40	LGB	Ketamine, Dexmedetomidine, Magnesium Sulfate,	Basal and post-surgery cytokine serum levels IL-1β, IL-6, and TNF-α	IL-1β in pre-surgery and post-surgery subjects, significant differences were found (49.58 pg/mL (18.50–112.20) vs. 13 pg/mL (5.43–22), respectively, *p* = 0.019)
Menck et al. [17]	2022	60	LGB	Ketamine, Dexmedetomidine, Lidocaine, Magnesium Sulfate	Pain scores, morphine consumption, delay in discharge from PACU, Rescue morphine	No statistically significant differences between the groups

**Table 2 healthcare-12-01094-t002:** Most frequently used coanalgesics and simple analgesics. NSAID—non-steroidal anti-inflammatory drug.

Agent	Mechanism of Action	Benefits	Potential Side Effects and Risks
Lidocaine i.v.	Blocks voltage-gated sodium channels, hyperpolarization-activated cyclic nucleotide channels, G protein-coupled receptors and potassium receptors, increases intracellular calcium concentration, blocks neutrophil priming	Analgesic, antihyperalgesic and anti-inflammatory properties	Bradycardia, hypotension, risk of toxicity
Ketamine	NMDA receptor antagonist	Analgesic, antihyperalgesic properties	An increase in systemic vascular resistance, tachycardia, hypertension, and risk of hallucinations, may affect bispectral index monitoring
S-Ketamine			
Magnesium Sulfate	NMDA receptor antagonist	Analgesic, antihyperalgesic and antiarrhytmic properties	Prolongation of nodal conduction times, PR, and QRS duration, risk of bradycardia, hypotension, augmenting muscle relaxation
Dexmedetomidine	Alpha 2 adrenergic receptor agonists	Analgesic, inhibiting the sympathetic outflow, antihyperalgesic, decreasing anesthetic requirement, anxiolysis, reduction of shivering threshold	Bradycardia, hypotension, potential vasoconstriction, a sedative effect
Clonidine			
Gabapentin	Calcium channel subunit alpha2-delta, gamma-aminobutyric acid analogs	Analgesic, antihyperalgesic, anxiolysis	A sedative effect, dizziness, blurred vision
Pregabalin			
Dexamethasone	Glucocorticoid	Anti-inflammatory, reducing pain scores after laparoscopy, preventing PONV	Hyperglycemia
Esmolol	Beta 1 adrenergic receptor antagonist	Maintaining hemodynamic stability, short-acting agent	Bradycardia, no analgesic effect
Labetalol	Beta 1,2 and alpha 1 receptor antagonist	Maintaining hemodynamic stability, vasodilatation	Hypotension, bradycardia, asthma exacerbation
Paracetamol	Prostaglandin synthesis inhibitor, possibly other mechanisms	Analgesic, antipyretic	Usually well-tolerated, liver dysfunction requires dose adjustment
NSAID	Inhibition of the cyclooxygenase enzymes	Analgesic, antipyretic, anti-inflammatory	Ulceration or bleeding from the gastrointestinal tract, kidney failure, coagulopathy, drug interactions
Metamizole	The precise mechanism is unknown	Analgesic, antipyretic, spasmolytic	Agranulocytosis, Anaphylaxis, potential for liver toxicity

## Data Availability

The data that support this study are available within the reference or available from the authors upon request.

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
