# Peer review of "Opioid-Free Anesthesia in Bariatric Surgery: Is It the One and Only? A Comprehensive Review of the Current Literature"

_healthcare, 2024, doi:10.3390/healthcare12111094_

Round 1

Reviewer 1 Report

Comments and Suggestions for Authors

This study concluded that the evidence supporting the use of opioid-free anesthesia as a standard technique for bariatric surgery is limited when compared to the effects of anesthesia with opioid use. These effects were assessed in terms of postoperative nausea and vomiting (PONV), pain management, opioid tolerance/induced hyperalgesia, and their impact on surgical nociceptive monitoring and respiratory/hemodynamic effects. The study is deemed clinically significant for its comprehensive literature review aimed at proposing an effective and safe anesthesia approach for bariatric surgery, particularly amidst increasing rates of obesity resulting from socioeconomic development.

 However, the opioid-free anesthesia-related papers reviewed in this study employed drugs with varying mechanisms of action, including alpha 2 adrenergic receptor agonist, lidocaine, ketamine, and magnesium. Consequently, due to the differing mechanisms of these drugs, the study's conclusions may vary depending on drug selection. It would be beneficial to categorize the analysis based on drugs with similar mechanisms of action and conduct a detailed literature review to ascertain any differences in effectiveness across the various parameters evaluated in this study

Author Response

Dear Reviewer,

Thank you for your thorough lecture on our manuscript. Below are our point-by-point responses to your comments.

Your comment: This study concluded that the evidence supporting the use of opioid-free anesthesia as a standard technique for bariatric surgery is limited when compared to the effects of anesthesia with opioid use. These effects were assessed in terms of postoperative nausea and vomiting (PONV), pain management, opioid tolerance/induced hyperalgesia, and their impact on surgical nociceptive monitoring and respiratory/hemodynamic effects. The study is deemed clinically significant for its comprehensive literature review aimed at proposing an effective and safe anesthesia approach for bariatric surgery, particularly amidst increasing rates of obesity resulting from socioeconomic development.

 However, the opioid-free anesthesia-related papers reviewed in this study employed drugs with varying mechanisms of action, including alpha 2 adrenergic receptor agonist, lidocaine, ketamine, and magnesium. Consequently, due to the differing mechanisms of these drugs, the study's conclusions may vary depending on drug selection. It would be beneficial to categorize the analysis based on drugs with similar mechanisms of action and conduct a detailed literature review to ascertain any differences in effectiveness across the various parameters evaluated in this study

Our response: Thank you for highlighting an important problem with the assessment of OFA for various surgical procedures. The literature reveals a vast heterogeneity of OFA protocols, including differences in drug selection and dosing regimen.  The influence of the isolated influence of most commonly used agents like dexmedetomidine or ketamine has been previously well described in meta-analyses and reviews. Still, as clinicians, we recognize the fact, that in practical application, OFA is a combination of drugs with different mechanisms of action that exist as a technique of its own. In spite of many differences in particular protocols, as you can see in the newly added Table 1, in almost all of the presented studies 3 or more co-analgesics were used. Therefore, from a practical standpoint, it is valuable to assess and demonstrate the impact of the OFA technique rather than focusing on a particular drug.

Taking into account your suggestions, we have outlined, where applicable, the possible impact of different drugs on assessed effects. Furthermore, Table 2 demonstrates the mechanisms and risks of co-analgesics divided by drug class. The above-mentioned justification for such a presentation of the OFA technique has also been included in the discussion section.

Thank you for your comments. We tried to implement them to improve the quality of our manuscript. We hope that the changes will be satisfactory for meeting the Healthcare publishing criteria. 

Yours sincerely,

Piotr Mieszczański

Reviewer 2 Report

Comments and Suggestions for Authors

Dear colleagues!

Issues of quality pain relief and opioid withdrawal are an important problem in modern medicine.

In general, your study is interestingly compiled, but I want to draw attention to the lack of the following data

1. Epidemiology of the use of opioids in the world as an anesthetic practice: in the introduction you do not pay attention to the prevalence of this type of anesthetic, and also do not write about adverse events as a result of administration

2. The article lacks sections: purpose, materials and methods, results

3. The article does not contain a null hypothesis, a procedure for searching literature sources, criteria for inclusion and exclusion from the list, in a word, the design of the review study was missed.

4. In section 9. Intraoperative nociception and monitoring from line 253, in my opinion, there is not enough data on pain tolerance, assessment of the effectiveness of pain relief and the patient’s quality of life

5. References

I have a question about a number of sources. For example, paragraph 2 (DOI: 10.1007/s11695-024-07118-3) contains only 7 authors in the original, but you mention more than 10; printed on pages 1075-1085, which you do not link to.

In this regard, I consider it correct to methodically and scrupulously double-check each item in the list of references in accordance with the original

You have sources that are more than 10 years old or more, while the materials and methods have no restrictions on search time.

Author Response

Dear Reviewer,

Thank you for your thorough lecture on our manuscript. Below are our point-by-point responses to your comments.

Your comment:  1 Epidemiology of the use of opioids in the world as an anesthetic practice: in the introduction you do not pay attention to the prevalence of this type of anesthetic, and also do not write about adverse events as a result of administration

Our response: We agree with the reviewer on this point. We have supplemented the introduction with an appropriate background.

Your comment:  2 The article lacks sections: purpose, materials and methods, results

Our response: The reviewer is correct; the structure of the manuscript has been corrected to meet the journal’s requirements for a review article.  

Your comment: 3 The article does not contain a null hypothesis, a procedure for searching literature sources, criteria for inclusion and exclusion from the list, in a word, the design of the review study was missed.

Our response: The null hypothesis, a procedure for searching sources, and criteria for inclusion and exclusion have been included in the purpose and material and methods sections to clearly demonstrate the study design and methodology used.

Your comment: 4 In section 9. Intraoperative nociception and monitoring from line 253, in my opinion, there is not enough data on pain tolerance, assessment of the effectiveness of pain relief and the patient’s quality of life

Our response: Intraoperative assessment of the patient's pain during general anesthesia is based on the analysis of the phenomenon of nociception and, potentially, stress response, although there are controversies about how to measure it and what is its clinical significance. Assessment of quality of life of a patient with pain, or assessment of the effectiveness of pain treatment is secondary to pain perception - which cannot be assessed in a patient undergoing general anesthesia. For this reason, and for the lack of scientific evidence in this field we did not develop these issues in the manuscript. Issues relating to postoperative pain treatment and long-term sequelae have been covered in sections 3.6 and 3.7. Still, we expanded the section to include the data to cover the immune and stress responses associated with OFA use.

Thank you for highlighting this important issue. To our knowledge, only one study assesses the long-term influence of OFA on chronic pain incidence in a relatively small group of patients and not in bariatric surgery (cited in the manuscript as [74] in section 3.7). This is indeed the direction of future research, which we also highlighted in the discussion.

Your comment: 5 References. I have a question about a number of sources. For example, paragraph 2 (DOI: 10.1007/s11695-024-07118-3) contains only 7 authors in the original, but you mention more than 10; printed on pages 1075-1085, which you do not link to.

In this regard, I consider it correct to methodically and scrupulously double-check each item in the list of references in accordance with the original

Our response: Thank you for pointing out this problem; the references have been proofread and, where necessary, corrected. 

Your comment: You have sources that are more than 10 years old or more, while the materials and methods have no restrictions on search time.

Our response: We decided against setting restrictions on search time. Absolute majority of randomized controlled trials studies on which we focus are less than 10 years. However, to give context and present the evidence, especially in field of drugs’ (both co-analgesics and opioids) properties but also pathophysiology of obesity, older sources had to be cited.

Thank you for your comments, we tried to implement them to improve the quality of our manuscript. We hope that changes will be satisfactory for meeting the Healthcare publishing criteria.

Yours sincerely,

Piotr Mieszczański

Reviewer 3 Report

Comments and Suggestions for Authors

interesting article.  just a couple comments.

1. The summary discusses disadvantages of OFA as well as the perioperative use of opioids.  It doesn't really cover the in between.  The idea of an opioid sparing anesthetic, where opioids are given judiciously and coanalgesics are utilized to achieve pain control and minimize opioid consumption.  Could the authors touch on this?

2. Could the authors comment on the role of neuraxial or regional techniques in an opioid free anesthetic?

Author Response

Dear Reviewer,

Thank you for your thorough lecture on our manuscript. Below are our point-by-point responses to your comments.

Your comment: 1 The summary discusses disadvantages of OFA as well as the perioperative use of opioids. It doesn't really cover the in between. The idea of an opioid sparing anesthetic, where opioids are given judiciously and coanalgesics are utilized to achieve pain control and minimize opioid consumption.  Could the authors touch on this?

Our response: Thank you for your comment; we altered the conclusions section to clarify that currently, anesthesia with the use of multimodal techniques to reduce the amounts of opioids given is considered to be a standard in bariatric anesthesia (according to ERAS guidelines). We admit that the “conventional anesthesia with opioid use” term that was present in the first version of the manuscript may be somewhat misleading, suggesting that liberal administration was an option in this patient’s group. Comparing OFA with opioid-sparing anesthesia with multimodal analgesia is a direction for future research, as we stated in the discussion section.

Your comment:  2 Could the authors comment on the role of neuraxial or regional techniques in an opioid free anesthetic?

Our response: We agree with the reviewer that regional anesthesia is an important element of OFA. We described it more widely in the 3.1 section of the manuscript. Moreover, we have added a comment on the potential role of neuraxial anesthesia in the same section with appropriate references.

Thank you for your comments, we tried to implement them to improve the quality of our manuscript. We hope that the implemented changes will be satisfactory for meeting the Healthcare publishing criteria.

Yours sincerely,

Piotr Mieszczański

Reviewer 4 Report

Comments and Suggestions for Authors

The authors approach a relevant and intriguing theme. Overall, the paper is well-written, but there is place for improvement in both structure and text:

  • The structure of the article does not match this journal's requirements for a review article: "All review papers should have the following structure: Abstract, Keywords, Introduction, Methods, Results, Discussion, and Conclusions."(Instructions for Authors). I strongly suggest a Methods/ Methodology section to reveal what databases the authors searched, the period, and the terms used in this search. I recommend a flow diagram about what type of articles you searched for, what you included in the study, and what you decided to reject from the analysis. The headings 2 to 9 could be included in the Results section.
  •  I suggest a Discussion section that may contain a glimpse of your personal experience,  and an opinion on the matter together with the controversies, the certainties, and the future directions of opioid-free anesthesia and analgesia. It should be concise.
  • There are no tables and figures in your article. That is unusual even for a review. I suggest a table with the relevant randomized trials: number of patients, pathology, primary outcomes, and results. You could also draw a figure showing the action mechanism of the drugs most used in OFA, but this is only a suggestion.  
  • cite "Articles reporting the complete elimination of opioids both during and after surgery [11,12] are based on single case reports and not on routine, reliable practice." Depending on the type of surgery, this statement may be exaggerated.

I conclude that the article is well-written, but it needs a substantial touch regarding its structure and the use of tables and figures. 

Author Response

Dear Reviewer,

Thank you for your thorough lecture on our manuscript. Below are our point-by-point responses to your comments.

Your comment: 

  • The structure of the article does not match this journal's requirements for a review article: "All review papers should have the following structure: Abstract, Keywords, Introduction, Methods, Results, Discussion, and Conclusions."(Instructions for Authors). I strongly suggest a Methods/ Methodology section to reveal what databases the authors searched, the period, and the terms used in this search. I recommend a flow diagram about what type of articles you searched for, what you included in the study, and what you decided to reject from the analysis. The headings 2 to 9 could be included in the Results section.

Our response: You are correct; the structure of the manuscript has been altered to meet the journal’s requirements, and the material and methods section and discussion have been added. Complying with your suggestions, we revealed what databases were searched, in what time period, and which keywords. A flow diagram of the selection process was added. The headings from 2 to 9 were included in the results to make the structure of the article more transparent.

Your comment: I suggest a Discussion section that may contain a glimpse of your personal experience,  and an opinion on the matter together with the controversies, the certainties, and the future directions of opioid-free anesthesia and analgesia. It should be concise.

Our response: Adhering to your suggestion, we have added the Discussion section concisely outlining our center’s experience, conflicting research, and certainties in the literature, as well as the proposed directions for future research.

Your comment: There are no tables and figures in your article. That is unusual even for a review. I suggest a table with the relevant randomized trials: number of patients, pathology, primary outcomes, and results. You could also draw a figure showing the action mechanism of the drugs most used in OFA, but this is only a suggestion.

Our response: We agree with the reviewer. We have added Table 1 with the relevant randomized trials, Table 2 with the mechanisms of action of the most frequently used co-analgesics and simple analgesics, and Figure 1, a flow diagram of the selection process.

Your comment: I cite "Articles reporting the complete elimination of opioids both during and after surgery [11,12] are based on single case reports and not on routine, reliable practice." Depending on the type of surgery, this statement may be exaggerated.

Our response: We thank the reviewer for pointing out this fact. We actually had only bariatric surgery in mind here. This has been added to the manuscript.

Thank you for your comments, we tried to implement them to improve the quality of our manuscript. We hope that implementing the recommended changes will be satisfactory for meeting the Healthcare publishing criteria. 

Yours sincerely,

Piotr Mieszczański

Round 2

Reviewer 2 Report

Comments and Suggestions for Authors

Dear colleagues!

Thank you for your work and revision: i have no questions and i'm happy to see good results.